# Sustainable Workplaces and Employee Well-Being: A Systematic Review of ESG-Linked Physical Activity Programs

**DOI:** 10.3390/healthcare13233146

**Published:** 2025-12-02

**Authors:** Hsuan Yu (Julie) Chen, Chin Yi (Fred) Fang

**Affiliations:** Graduate Institute of Sport, Leisure and Hospitality Management, National Taiwan Normal University, No. 162, Section 1, Heping East Road, Da’an District, Taipei City 1068, Taiwan; julliechen333@gmail.com

**Keywords:** ESG-S, mental health promotion, occupational health culture, hybrid physical activity intervention

## Abstract

**Background**: Despite evidence of potential benefits, variability in exercise types, psychological outcomes, and methods hinders comprehensive cost-effectiveness evaluation, framed through Stimulus–Organism–Response (S–O–R) theory. In this context, Workplace Physical Activity-Based Programs (WPABPs) serve as environmental stimulation that influences employees’ emotional states, which in turn shape mental health outcomes and behavioral responses. **Research Purpose:** This systematic review examines WPABPs through the social dimension of the Environmental, Social, Governance (ESG-S) framework, with a focus on their impact on employees’ mental health. **Methods**: Following the Preferred Reporting Items for Systematic Reviews and Meta-Analyses (PRISMA) 2020 guidelines, eligibility was assessed via the PICO (Population, Intervention, Comparison, Outcome) framework. The ScienceDirect, Scopus, Google Scholar, and PubMed databases were searched using Medical Subject Headings (MeSH) aligned keywords and Boolean operators. **Results**: Of the 961 articles identified, 15 studies (2021–2025) met the inclusion criteria. WPABPs were found to improve employee mental health, reduce stress, and enhance well-being. Individualized interventions supported targeted psychological benefits, while group formats promoted social cohesion and engagement. Variations in type, duration, and delivery, as well as accessibility barriers for underrepresented employees, were noted. WPABPs enhance employee well-being and organizational outcomes, contributing to the Sustainable Development Goals (SDGs), specifically SDG 3 (Good Health and Well-being) and SDG 8 (Decent Work and Economic Growth). **Conclusions**: Hybrid models combining individual and group approaches with managerial and digital support are recommended. Integrating WPABPs within ESG-S and Corporate Social Responsibility (CSR) frameworks can institutionalize sustainable workplace health promotion, while future research should focus on standardized, inclusive, and long-term evaluations.

## 1. Introduction

### 1.1. Global Trends in Workplace Health Promotion

The World Health Organization’s (WHO) Global Action Plan on Physical Activity 2018–2030 urges governments and businesses to promote employees’ physical activity [1]. However, the WHO’s 2024 report reveals that 31% of adults worldwide failed to meet recommended physical activity levels in 2022, making the 2030 global goal challenging [2]. Meanwhile, the role of sport in achieving Sustainable Development Goals (SDGs) has gained international recognition [3]. The American College of Sports Medicine (ACSM) ranked workplace health promotion as 2024s top fitness trend, with exercise for mental health ranking eighth in 2024 and 2025 [4,5]. Work is a key social determinant of mental health. Evidence indicates that adverse working conditions and organizational factors (e.g., social inequalities) influence depressive symptoms and psychological distress [6]. Workplace mental health should be regarded as an integral component of management and labor practice organization [7].

### 1.2. Personal Psychological-Level Benefits and Work-Related Outcomes of WPABPs

As the global problem of insufficient physical activity becomes more severe, workplaces have been identified as critical settings for promoting regular physical activity among adults and improving mental health. Currently, there is no universally accepted or standardized definition of Workplace Physical Activity-Based Programs (WPABPs) in the international literature [8]. Previous scholars have demonstrated the potential of Physical activity (PA) as a multifaceted workplace intervention with a high degree of individualization of parameters for different employee groups and different types of workplaces to increase PA [9]. Past scholars have also defined “workplace interventions for increasing physical activity” as those interventions implemented within the workplace and designed to enhance employees’ levels of physical activity, including behavioral education, environmental modifications, coaching, or digital tools [10]. In this study, WPABPs are defined as systematically planned, periodic workplace physical activity courses by enterprises through the provision of qualified instructors and exercise facilities.

First, at the personal psychological level, previous research indicates that workplace physical activity programs can improve occupational health and have positive psychological, physiological, and social effects, particularly on parameters related to well-being and interpersonal relationships [11]. To encourage participation, employees who use activity-tracking apps with features such as team competitions, social sharing, and personal milestones have significantly increased their physical activity levels, health outcomes, and satisfaction [12]. Maintaining regular exercise can enhance physical functioning, buffer the effects of stress on mental and physical health, and promote a healthy social life at work, resulting in improved physical, psychological, and social well-being [13]. Second, at the level of work-related outcomes, studies have shown that employees who spend some work time on fitness can maintain or even improve productivity, leading to increased overall output [14]. Companies that prioritize employee well-being often report better job satisfaction and overall performance [15]. Additionally, employee well-being’s unique, lasting, and far-reaching benefits help businesses remain competitive in the marketplace [16]. In today’s high-turnover society, satisfied employees are more likely to become loyal advocates for their organizations [17].

In summary, the WPABPs may not only help enhance employees’ physical and mental health and well-being but also promote productivity, job satisfaction, and organizational loyalty. By combining individual health promotion and work performance improvement, physical activity programs have become important strategies for strengthening employee well-being and building sustainable competitive advantages for companies.

### 1.3. Need for Systematic Synthesis and Evidence Clarity

As workplace health promotion becomes increasingly widespread and systematized, internal environments for physical activity within organizations have become more supportive. However, existing research on WPABPs demonstrates significant diversity in intervention types and designs—ranging from fitness training, aerobic exercise programs, and strength training to muscle relaxation, walking, lifestyle interventions, resistance training, stress and time management, and cycling [18]. These interventions vary widely in intensity, duration, and participation patterns, making it difficult to compare or synthesize results across studies [19], which weakens the overall consistency and practical applicability of the literature.

Moreover, the outcome variables and evaluation indicators used in these studies are highly heterogeneous. These include, but are not limited to, mental health, stress, burnout, depression, anxiety, and sleep [20], and are often assessed without a unified theoretical framework or standardized measurement tools. Such variability makes it challenging to identify which types or designs of physical activity interventions are genuinely effective in promoting mental health [21].

Therefore, a systematic synthesis of the existing evidence is needed to clarify which types and conditions of workplace physical activity interventions can most effectively deliver dual benefits: enhancing employee mental health and advancing the social impact of the Environmental, Social, Governance (ESG) commitments.

### 1.4. Theoretical Mechanisms

#### 1.4.1. ESG

The ESG originated from the United Nations’ 2004 report Who Cares Wins, which reported that ESG represents key factors for assessing the sustainability and ethical impact of corporate investments. Effective management of these social issues can enhance corporate reputation and trust, attract and retain talent, and ensure legal and sustainable operations [22]. ESG can be regarded as a more quantifiable extension of Corporate Social Responsibility (CSR) within financial economics research, focusing specifically on environmental, social, and governance performance [23]. The Environmental dimension concerns corporate responsibility toward the natural environment and resource sustainability; the Social dimension focuses on responsibility and relationships with people and society; and the Governance dimension emphasizes internal mechanisms that ensure accountability, transparency, and long-term value creation [24].

Prior research has shown that workplace diversity, inclusion, and employee development positively influence banks’ overall and social ESG performance [25]. Enhanced ESG awareness increases work meaning and psychological well-being. Moreover, as pay satisfaction rises, work meaning strengthens the mediating role between perceived ESG and well-being [26]. Transparency in environmental information and effective internal control mechanisms are also key drivers of employee satisfaction, with top managers’ environmental awareness and employees’ education levels reinforcing this relationship [27]. In terms of corporate social activities, improvements in job satisfaction and work–life balance are associated with reduced occupational stress, while social initiatives can enhance employees’ psychological health [28].

In summary, this study explores the impact of the WPABPs on employees’ psychological well-being from the ESG Social perspective.

#### 1.4.2. S–O–R Theory and Applications

The Stimulus–Organism–Response (S–O–R) theory developed by Mehrabian and Russell (1974) [29] interprets how environmental stimuli influence an individual’s internal conditions (Organism), which subsequently drive behavioral outcomes (Response). External factors stimulate both emotional and cognitive mechanisms that shape human actions.

Table 1 summarizes six empirical studies that apply the S–O–R framework to workplace employees. It outlines how each study defines the three components: stimuli, organism, and response and how the variables are operationalized. The included studies span various workplace contexts, such as green management, human resource practices, smart technology adoption, organizational policies, and corporate social responsibility. By comparing the sources of stimuli, employees’ internal psychological processes, and resulting behavioral or attitudinal outcomes, the table highlights both the diversity and common patterns in S–O–R applications across workplace topics. It also provides a useful reference for future researchers developing S–O–R-based models.

Prior research has indicated that environmentally focused transformational leaders in eco-friendly hotels enhance staff’s emotional dedication to sustainability goals. Other investigations have shown that such leadership styles encourage innovation and teamwork, ultimately improving productivity. Moreover, heightened stress as a stimulus increases burnout (O), which consequently reduces both job and life satisfaction (R). Extending these findings, recent S–O–R-based research further demonstrates that green hotel practices stimulate employees’ internalized environmental motivation, leading to stronger pro-environmental behaviors [30]. In human resource management contexts, supportive HRM stimuli strengthen employees’ psychological states, enhancing motivation and productivity [31]. Similarly, workplace conditions as stimuli play a crucial role in shaping employees’ job and life satisfaction through their internal evaluations [32]. In logistics settings, smart technology functions as a stimulus that improves employees’ well-being and learning performance via enhanced cognitive-emotional states [33]. Organizational energy-saving cues also activate employees’ environmental awareness and perceived responsibility, resulting in greater energy-saving behavior [34]. Furthermore, CSR initiatives act as positive organizational stimuli that elevate employees’ subjective well-being, subsequently promoting creative behavior and lowering turnover intention (Chung et al., 2024) [35]. Across the six studies, S–O–R theory consistently demonstrates that workplace stimuli—ranging from organizational policies, CSR initiatives, HRM practices, environmental cues, to smart technologies—shape employees’ internal psychological states, which in turn influence key behavioral and attitudinal outcomes. Despite variations in contexts and operationalization, the studies collectively show that positive and supportive stimuli tend to enhance employees’ well-being, motivation, environmental consciousness, and satisfaction, ultimately promoting desirable responses such as creative behavior, green behavior, energy-saving practices, improved performance, and reduced turnover intention. This synthesis highlights the robustness and adaptability of the S–O–R framework for explaining employee cognition, emotion, and behavior across diverse organizational settings.

WPABPs can be interpreted through the S–O–R framework, which explains stimuli (WPABPs) that influence the organism (employees’ mental health), ultimately shaping the response (employee behavior). Potential confounding arises when external factors moderate the organism’s response. Mediating mechanisms include positive affect, psychological need satisfaction, and perceived social support, which explain how WPABPs produce outcomes beyond mere physical health improvements.

### 1.5. Research Objectives

This systematic literature review investigates how WPABPs operate as a strategic mechanism under the social dimension of the ESG-S framework, focusing on their role in enhancing employees’ mental health and supporting sustainable organizational development. The study hypothesizes that regular participation in workplace physical activity and related interventions positively affect mental health by reducing stress and depression risks, improving psychological well-being, and increasing job satisfaction. These outcomes, in turn, may enhance overall work performance and foster healthier, more equitable workplace environments. All of which are consistent with the ESG-S goals of promoting a healthier workforce.

Specifically, this review aims to (1) classify and summarize key types and implementation modes of WPABPs, (2) analyze the associations between various physical activity interventions and psychological health outcomes, (3) examine their contributions to ESG-driven social practices in corporate contexts, and (4) propose an integrative framework to guide future research and practice. Through this synthesis, the study offers both theoretical insights and practical implications for organizations, policymakers, and leaders seeking to align corporate health initiatives with societal well-being.

### 1.6. The Significance of Research and Review Scope

The significance of this study lies in addressing the growing global emphasis on Workplace Health Promotion as a key strategy for achieving sustainable development and employee well-being [4]. WPABPs have emerged as a central component of WHP, supported by the WHO and international health guidelines, which emphasize the need to reduce sedentary behavior and promote regular physical activity to achieve the United Nations SDGs [3].

However, existing research reveals considerable heterogeneity in intervention types, intensity, duration, and evaluation methods, making it difficult to determine which program designs most effectively enhance both employee mental health and organizational performance [19]. Moreover, variations in workplace culture and resources create challenges for implementation feasibility and sustainability. Accordingly, the significance of this study can be summarized as follows:Responding to global trends in workplace health promotion by systematically synthesizing empirical evidence on WPABPs to clarify their integrated effects on mental health and work-related outcomes.Highlighting the linkage with the ESG-S, examining how WPABPs contribute to sustainable business practices.Applying the S–O–R theoretical framework to explain how WPABPs function as external stimuli that influence employees’ internal psychological states and behavioral responses.Bridging the gap between research methods and practical application to guide future interventions aimed at reducing workplace health inequities.

Regarding the research scope, this study included a total of 15 reviewed articles (participants = 361,165). The primary inclusion criteria focused on workplace health promotion as the central theme and employees as the target population.

## 2. Methods

This study followed 2020s Preferred Reporting Items for Systematic Reviews and Meta-Analyses (PRISMA) guidelines to ensure methodological rigor in research question formulation and review process [36].

### 2.1. Database Selection and Source Justification

A database search was conducted using PubMed, Scopus, Google Scholar, and ScienceDirect. PubMed is a free biomedical database with precise Medical Subject Headings (MeSH) indexing and high-quality peer-reviewed content. Scopus covers science, technology, medicine, social sciences, and humanities. Google Scholar provides broad coverage, indexing books, theses, gray literature, and non-peer-reviewed materials with citation metrics [37]. ScienceDirect offers full-text access to journals in medicine, social sciences, and humanities [38].

### 2.2. Keyword Strategy and Boolean Search Logic

Keywords were selected and combined using Boolean operators (AND, OR, NOT) to define the search logic. This allowed the review to broaden or narrow the literature scope and achieve standardized, replicable search results (Table 2).

### 2.3. Screening and Methodological Quality Assessment

Initial screening followed the PICO (Population, Intervention, Comparison, Outcome) framework and applied the inclusion and exclusion criteria detailed in Table 3. This review included only studies with full-text availability to ensure complete assessment of methodology and results, thereby avoiding bias caused by incomplete information. Additional criteria included retaining only non-duplicate records, full-text articles, English-language publications, and studies published within the past five years. These decisions aimed to enhance methodological consistency, comparability, and the overall reliability of the synthesized evidence. Methodological rigor served as a primary assessment criterion throughout the process. Studies lacking clearly defined designs, methodological transparency, or consistent findings were excluded to ensure analytical integrity and data reliability. Experimental abstracts, case reports, editorials, and letters to the editor were also removed. Systematic searches across electronic databases were then conducted, supplemented by abstract tracking. Eligible studies were synthesized to identify knowledge gaps and underexplored issues, thereby enhancing the coherence, validity, and explanatory strength of the review.

The results of this systematic literature review were compiled from a qualitative synthesis perspective, focusing on the research methods and structural design of each included study. The entire analytical process was conducted according to the PRISMA 2020 guidelines, ensuring a rigorous evaluation and synthesis.

### 2.4. Three-Stage Literature Search Strategy

The literature search process was divided into three main stages.

Stage 1: Preliminary Screening

A broad search of key data sources was performed. Studies deemed unsuitable, such as experimental abstracts, case reports, editorials, and letters to the editor, were excluded to establish a clear research direction. Duplicate records were also identified and removed to ensure the uniqueness of retrieved studies.

Stage 2: Systematic Database Search

Highly relevant studies were targeted through systematic searches of electronic databases, complemented by abstract tracking to broaden coverage. All remaining articles were read in full, with careful verification of their alignment with the inclusion criteria and consistency across objectives, methods, and results.

Stage 3: Synthesis and Validation

The collected information was synthesized and systematically organized to identify knowledge gaps and underexplored issues in the literature. The PRISMA flowchart was thoroughly reviewed to confirm numerical accuracy, clearly report exclusion counts at each stage, and provide explicit reasons for exclusion. These procedures were implemented to minimize bias and strengthen the overall credibility of this systematic review.

### 2.5. Reference Management and Initial Search Results by Database

The preliminary search retrieved 961 articles: 14 from PubMed, 49 from Scopus, 223 from Google Scholar, and 675 from ScienceDirect. After completing the standardized selection procedures, a total of 15 articles met the inclusion criteria. A detailed breakdown of the final included articles by database is presented in Table 4 as the foundation for subsequent data analysis. EndNote 21 facilitated literature management, annotation, and citation formatting.

A total of 961 records were initially identified through the database search. After the first phase of removing duplicates, 956 records remained and were screened based on full-text availability, publication language, and year. Following the second phase of exclusion, 180 records were considered potentially relevant and were retrieved for full-text review. In the third round of screening, an additional 165 articles were excluded for not meeting the inclusion criteria, primarily for the following reasons: the study was not related to occupational health promotion, or the target population did not consist of employed individuals. Ultimately, 15 studies met the inclusion criteria. The PRISMA flowchart (Figure 1) illustrates the screening process, enhancing the accuracy and visualization of the overall review process.

For systematic literature reviews guided by the PRISMA framework, the literature search and selection processes are systematically structured, even when a formal risk-of-bias assessment is not applied. Clear reporting of search strategy, databases, time frame, and inclusion/exclusion criteria ensures methodological transparency and replicability [39,40].

## 3. Results

The selected articles in this review were systematically categorized and summarized in Table 5 based on the following dimensions: Author, Year, Title, Intervention Intensity (including program duration), Type, Participants, Country and Industry (e.g., the private, public, and voluntary sector, educational institutions, or the financial industry, etc.), and Key Results. Overall, WPABPs effectively enhance multiple employee and organizational outcomes. However, considerable heterogeneity exists in program design and implementation across different occupational and national contexts. A wide range of intervention types and intensities was identified, including moderate-to-vigorous aerobic exercise, yoga-based programs, behavioral regulation and smartphone-use reduction interventions, individualized and trainer-led exercise programs, and technology-supported initiatives. For example, reducing smartphone use while increasing daily physical activity significantly improved job satisfaction, motivation, work–life balance, and mental health among employees in Germany [41]. Similarly, allowing employees one paid hour per week for physical activity led to higher activity levels, improved mental health, and greater job satisfaction among healthcare staff [42].

Systematic reviews further support these findings. Physical Activity (PA-based) corporate programs across Europe, Australia, and the USA demonstrated health improvements and short-term economic returns [43], while worksite wellness programs across multiple countries enhanced both worker health and productivity [19]. Studies in developing contexts echoed these outcomes, showing that wellness programs were significantly associated with improved employee performance in Tanzania’s banking sector [44], and that regular movement in office environments improved general well-being and cognitive functioning [45]. Meanwhile, evidence from the Philippine Business Process Outsourcing (BPO) industry underscored elevated risks of physical and psychological strain among employees, highlighting the urgent need for structured workplace health promotion [46].

Program context and implementation conditions emerged as critical determinants of success. Long-term yoga interventions for desk-based and remote workers enhanced stress relief, sleep quality, and life satisfaction [47], while short team-based PA sessions improved engagement and well-being, although time constraints and motivational barriers persisted [12]. Managerial support and organizational culture significantly influenced PA behaviors and mental health outcomes [21].

Interpersonal and psychosocial factors were also shown to play key mediating roles. Alignment between trainers’ supportive behaviors and exercisers’ psychological needs strengthened intrinsic motivation and well-being in fitness settings [48], and personalized, trainer-led exercise programs enhanced mood and workplace atmosphere among university staff [49]. Group-based exercise and positive affect promoted reciprocal improvements in emotional well-being and participation consistency [50]. In addition, regular workplace exercise programs were shown to reduce musculoskeletal disorders, occupational stress, and burnout, fostering healthier lifestyles and potential economic benefits [51,52].

Overall, the collective evidence confirms that WPABPs effectively promote employee mental health, work-related outcomes, and organizational sustainability. Program success depends on the interplay of intervention intensity, managerial and social support, accessibility, and long-term engagement. Integrating these components can maximize both employees’ mental health and organizational benefits, and sustainable corporate performance within the ESG-S dimension.

**Table 5 healthcare-13-03146-t005:** Identification and Inclusion of Articles for the Review.

Author	Year	Title	Intervention Intensity	Type	Participants	Country and Industry	Results
Brailovskaia et al. [41]	2024	Less smartphone and more physical activity for a better work satisfaction, motivation, work-life balance, and mental health: an experimental intervention study	Moderate-intensity aerobic and behavioral regulation program, 6-week intervention, 30 min/day increase	Test combining PA increase and smartphone-use control	Employees across different professional sectors and workplaces (n = 278)	Germany, mixed workplaces (private and public; multiple industries)	Reduced smartphone use combined with PA interventions significantly improved job satisfaction, motivation, work–life balance, and positive mental health, while reducing work overload and problematic phone use.
Candelario et al. [46]	2024	Integrative review of workplace health promotion in the business process outsourcing industry: focus on the Philippines	Varied (incorporating different intensities, durations, types, and frequencies of exercise)	Integrative review (n = 37 studies)	Target population: BPO workers described in included studies (n = 261,262)	The Philippines, the BPO industry (private sector)	BPO workers face elevated risks of physical and psychological strain, sleep disorders, and occupational diseases due to the inherent demands of their work.
Msuya & Kumar [44]	2022	Nexus between workplace health and employee wellness programs and employee performance	Not reported (survey)	Cross-sectional/ survey study	Banking employees (n = 252)	Tanzania, Financial industry (banking), private/public banking sector	A significant positive relationship was found between workplace health and wellness programs and employee job performance.
Paramashiva et al. [47]	2025	Enhancing well-being at work: qualitative insights into challenges and benefits of long-term yoga programs for desk-based workers	Varied (yoga intensity moderate/low; long-term programs)	Semi-structured interviews with a long-term yoga program	Participants in a long-term workplace yoga program (desk-based or remote workers, n = 6)	India, Division of Yoga, Manipal Academy (educational/research setting)	Among remote workers, strong correlations were observed between attendance, daily stress, stress relief, sleep quality, work purpose, and life satisfaction. Remote work stress and sleep quality were strongly related to stress relief and work purpose.
Singh et al. [12]	2024	Evaluation of the “15-min challenge”: a workplace health and wellbeing program	15 min daily PA sessions (light–moderate) for six weeks.	mHealth gamified program (team competition and activity logging)	Employees from multiple workplaces participating in the “15-Minute Challenge” (n = 73)	Australia, New Zealand, UK, mixed workplaces (mostly private/organizational settings)	Yoga participants reported challenges maintaining regular practice due to time constraints, initial discomfort, and varying motivation, yet noted substantial benefits in mental health and stress management.
Larisch et al. [21]	2023	“It depends on the boss”—a qualitative study of multi-level interventions aiming at office workers’ movement behaviour and mental health	During the six-month intervention period, (1) focused on increasing moderate-to-vigorous PA, (2) targeted replacing sedentary behavior with low-intensity PA.	Qualitative process evaluation of multi-level RCT interventions	Office workers (n = 38)	Sweden, office/white-collar workplaces (private and public offices)	The “15—Minute Challenge” effectively increased employee PA levels and improved self-reported health outcomes. High satisfaction and significant health improvements highlight the potential of workplace wellness programs to counter sedentary behavior and promote active lifestyles.
Bonatesta et al. [43]	2024	Short-term economic evaluation of physical activity-based corporate health programs: a systematic review	Varied (incorporating different intensities, durations, types, and frequencies of exercise)	Systematic review (n = 11 studies)	Multiple PA–based corporate health programs (n = 60,020)	Europe, Australia, and USA, corporate programs (predominantly private sector)	Office employees attributed improvements in movement behavior (particularly reduced sedentary) and well-being to the interventions, notably cognitive-behavioral therapy–based counseling and free gym access. Managerial and team support were identified as key facilitators.
Gil-Beltrán et al. [51]	2020	Get vigorous with physical exercise and improve your well-being at work!	Vigorous exercise emphasized (higher intensity)	Intervention/commentary promoting vigorous PA	Workplace samples (n = 485)	Spain and Latin America, private companies	Beyond fostering healthy lifestyles, PA-based corporate health programs demonstrate potential for generating substantial short-term economic returns.
Marin-Farrona et al. [19]	2023	Effectiveness of worksite wellness programs based on physical activity to improve workers’ health and productivity: a systematic review	Varied (Most RCTs ranged from 6 to over 12 weeks, incorporating different intensities, durations, types, and frequencies of exercise)	Systematic review of worksite PA programs (n = 16 studies)	Samples included healthcare workers, office and industrial workers, university staff, cleaners, postal workers (n = 36,623)	Multiple countries (Denmark, Norway, UK, Brazil, Germany, USA, Spain, Japan, The Netherlands, parts of Africa), mixed: healthcare (public), industrial (private), educational institutions, services	Operability was identified as the productivity variable most influenced by worksite programs based on PA, effectively enhancing both worker productivity and health.
Ford et al. [42]	2022	Impacts of a workplace physical activity intervention on employee physical activity & mental health for NHS staff in Wales: an evaluation of the pilot time to move initiative	Offers employees a 12-month program allowing one hour per week of paid time for participation in chosen physical activities.	Organizational initiative (Time to Move initiative; workplace PA program)	NHS staff in Wales (n= 625)	Wales is one of the four public sector healthcare systems under the UK	Providing paid time for PA yielded positive outcomes for many employees, including higher activity levels, improved mental health, and greater job satisfaction.
Munuo [45]	2023	Physical activity at work and job performance: a qualitative study of physical activity at work from the perspective of office workers in Tanzania.	At least 150–300 min of moderate-intensity aerobic PA per week. Interview: (a) prerequisites to be physically active at work (b) barriers to PA at work (c) how PA influences their job performance at work.	A qualitative study	Banking managers, senior and junior officers (n= 9)	Tanzania (Shinyanga Region), Financial industry (banking)	Regular movement in office settings contributed to better employee health and overall job performance, while workplace health promotion activities enhanced general well-being and cognitive functioning through exercise.
Martinez [52]	2021	The importance of workplace exercise	Varied (incorporating different intensities, durations, types, and frequencies of exercise)	Narrative review (n = 14 studies)	Workplace employees across various occupations	Brazil, Salvador (and other regions cited), mixed workplaces (private/organizational settings)	Workplace exercise directly improves employees’ quality of life, reducing the incidence of repetitive strain injuries, work-related musculoskeletal disorders, occupational stress, and burnout syndrome.
Rodrigues et al. [48]	2021	Trainer-exerciser relationship: the congruency effect on exerciser psychological needs using response surface analysis	Not reported (observational: looks at perceived interpersonal behavior quality)	Cross-sectional survey of trainer–exerciser interactions	Coaches (n = 130), gym exercisers (n = 640)	Portugal, private gyms/fitness industry (private sector)	Coaches tended to overreport supportive and underreport thwarting behaviors; alignment with exercisers’ perceptions improved basic need satisfaction, particularly regarding relatedness.
Casimiro-Andújar et al. [49]	2022	Effects of a personalised physical exercise program on university workers overall well-being: “UAL-Activa” program	Sessions are tailored to individual goals and fitness levels, conducted once per week for six months, each lasting 60 min. Including strength training, aerobics exercises etc.	Personal training/ individualized exercise program at a gym outside of work	University workers participating in a personalized exercise program (n = 25)	Spain, educational institution (university of Almeria staff; public university)	Participation in personalized, trainer-led exercise programs improved overall health and mood, producing a notably positive impact on the work environment.
Gil-Beltrán et al. [50]	2024	How physical exercise with others and prioritizing positivity contribute to (work) wellbeing: a cross-sectional and diary multilevel study	Exercise was measured by weekly frequency, session duration, and intensity (0–5 scale).	Cross-sectional + diary multilevel study; group exercise settings	Study 1 (n = 553) and Study 2 (n = 146) focused on group exercise during confinement.	Spain, mixed (community/worker samples; private and public settings)	Higher engagement and positive affect promoted greater exercise participation, particularly in positive social contexts. Prioritizing positive emotions also preceded and reinforced positive affect during exercise, creating a reciprocal relationship between emotion and physical activity.

Note: Physical Activity (PA); Business Process Outsourcing (BPO) industry; National Health Service (NHS).

## 4. Discussion

This systematic review integrated 15 empirical studies and clearly demonstrated that WPABPs as a stimulus (S) can effectively improve employees’ mental health (O) by increasing levels of physical activity. Further promote positive behavioral outcomes (e.g., exercise participation, productivity, and job performance), reduce negative behaviors (e.g., sedentary tendencies and perceived work overload) (R).

### 4.1. Overall Impact of WPABPs on Employees’ Physical and Mental Health

WPABPs have demonstrated substantial benefits for employees’ physical and mental health. Supportive interpersonal behaviors from coaches enhance exercisers’ autonomy, competence, and relatedness, promoting mental health and reducing emotional exhaustion [48]. Personalized, trainer-led programs improve overall health, mood, and engagement, with positive affect reinforcing exercise behavior [49,50].

Combining PA with mindful management of digital device use reduces depressive symptoms, enhances control, and improves psychological well-being [41]. Yoga and other individualized interventions improve stress management and mental health, despite challenges such as time constraints or initial discomfort [12]. Programs such as the “15-Minute Challenge” and interventions with managerial and team support increase PA levels, reduce sedentary behavior, and improve self-reported health outcomes [21,43]. Allocating paid time for exercise further enhances employees’ mental well-being [42].

WPABPs also address occupational stressors and sleep issues, particularly among Business Process Outsourcing (BPO) companies and remote workers, enhancing stress relief, sleep quality, work purpose, and overall life satisfaction [46,47]. Regular workplace exercise reduces musculoskeletal disorders, repetitive strain injuries, occupational stress, and burnout, while improving vitality, cognitive function, and overall well-being [45,52].

### 4.2. WPABPs and Work-Related Outcomes

WPABPs also generate positive work-related outcomes. Reduced smartphone use combined with exercise improves job satisfaction, motivation, work–life balance, and positive mental health while decreasing work overload [41]. Participation in structured PA programs, such as the “15-Minute Challenge,” personalized coaching, and cognitive-behavioral therapy–based counseling, improves movement behaviors, adherence, and overall work environment quality [21,43,49].

PA-based corporate health programs are associated with higher productivity, better job performance, and economic benefits [19,51]. Providing paid time for exercise further increases activity levels, mental health, and job satisfaction [42]. Engagement and positive affect during PA reinforce participation and enhance social cohesion, supporting organizational culture and work performance [50,53]. Overall, WPABPs contribute to enhanced employee efficiency, well-being, and workplace functioning, making them a strategic organizational tool.

### 4.3. Differential Impact of Individual vs. Group Interventions

In the field of workplace health promotion, both individual and group interventions are often implemented in parallel within WPABPs, yet each operates through distinct mechanisms and yields different effects on employee behavior, health outcomes, and organizational benefits [54].

Individual interventions typically involve trainer-led programs or self-directed participation tailored to employees’ goals, fitness levels, or motivational states. Evidence from several studies highlights that such interventions produce significant psychological and emotional benefits, including stress reduction, improved mood, greater job satisfaction, and enhanced work engagement. For example, university workers participating in the personalized “UAL-Activa” program reported overall improvements in health and well-being [49], while remote desk-based workers engaged in long-term yoga programs experienced reduced daily stress, better sleep quality, and increased sense of work purpose [47]. Similarly, individualized trainer–exerciser alignment was found to enhance basic psychological needs satisfaction, particularly when relatedness support was high [48]. Programs allowing flexibility for individual participation, such as the Time to Move initiative [42] or the mHealth gamified 15 min challenge [12], were also associated with higher adherence, engagement, and mental health benefits. Reviews in specific sectors, such as BPO industries, further highlight that individualized approaches are important to address elevated physical and psychological risks among workers [46]. These findings suggest that individualized interventions are particularly effective for addressing personal stressors, high-pressure roles, and targeted health outcomes, although they often require greater time, professional support, and resources.

In contrast, group-based interventions emphasize social interaction, collective engagement, and shared activities, fostering team cohesion, workplace culture, and motivation. Participation in group exercise settings has been shown to enhance consistency and enjoyment of physical activity, particularly when positive social contexts are emphasized [50]. Multi-level interventions targeting office workers’ movement and sedentary behavior [21] and corporate physical activity programs integrating team support and managerial involvement [43] demonstrated improvements in movement behavior, well-being, and self-reported health outcomes, while benefiting from economies of scale and broader organizational reach. Beyond these, evidence also indicates that vigorous physical activity programs can generate substantial short-term economic returns and promote healthy workplace cultures [51]. Group-oriented programs contribute to reducing musculoskeletal complaints, occupational stress, and burnout, while promoting social cohesion and collective engagement in health-promoting behaviors [52]. However, these interventions may be less effective in addressing individual differences, motivational barriers, or personal constraints, as the structure and intensity are often standardized across participants.

Some interventions integrate both approaches or do not explicitly differentiate between individual and group formats. For instance, programs combining physical activity with behavioral regulation or smartphone-use reduction [41], multi-site wellness programs [19], and qualitative studies of workplace activity patterns [44,45] indicate that blending individual flexibility with collective engagement can enhance adherence, motivation, and overall program effectiveness.

Additionally, differences in resource requirements and support systems distinguish individual and group interventions. Individual programs often demand greater professional support, personalized guidance, and time investment, making access to trained staff and organizational backing crucial [46,49]. In contrast, group interventions benefit from shared resources, peer motivation, and managerial support, which enhance scalability and collective engagement, though they may be less able to address individual needs [21,43,50].

Overall, individual and group interventions complement each other: individualized programs are particularly suited for achieving deep psychological gains, stress reduction, and targeted health outcomes, while group interventions promote social support, teamwork, and organizational culture, benefiting scalability and collective engagement. Optimal workplace health promotion strategies may therefore combine both approaches, tailoring interventions to employee needs, organizational resources, and cultural context to maximize both individual and organizational outcomes.

### 4.4. The Value of WPABPs Within the ESG-S Framework

The findings of this review indicate that implementing WPABPs not only improves employee health but also aligns with Corporate Social Responsibility (CSR) principles [46], and CSR structures critically influence how ESG strategies affect overall corporate performance [55]. Providing paid time for physical activity has long-term positive impacts for employees and strengthens a company’s caring image [42]. Consistent interpersonal evaluations between coaches and exercisers help establish a culture of care and respect, which is important for building a healthy workplace culture and realizing CSR [48]. Specifically, physical activity supports sustainable development and directly contributes to SDG 3: Good Health and Well-being, SDG 5: Gender Equality, and SDG 8: Decent Work and Economic Growth [38].

Aligning WPABPs with ESG-S initiatives, CSR efforts, and the United Nations SDGs, particularly SDG 3 on ensuring healthy lives, embeds them as sustainable organizational strategies [45,51,52]. By promoting physical and mental health, social cohesion, and workplace well-being, transform wellness programs from isolated initiatives into integrated, strategic interventions.

Therefore, WPABPs can serve as a practical, measurable, and high-impact tool for implementing the ESG-S strategies, enhancing organizational incentives and the sustainability of these programs. However, if companies view WPABPs only as short-term PR strategies rather than deep cultural changes, this may limit their long-term social value [47].

### 4.5. Intervention Heterogeneity and Future Implementation Challenges

Although WPABPs generally yield positive outcomes, significant heterogeneity exists in intervention design, including exercise type, intensity, duration, delivery mode, and individual employee differences [19]. Organizational context, such as company culture, departmental differences, and work environment, further contributes to variability [45,56]. Many studies also inadequately report cost-effectiveness data, limiting comprehensive economic evaluations [21]. Personalized interventions and managerial support are critical for maximizing effectiveness, yet they often demand substantial time, resources, and professional input, posing feasibility challenges, especially for small and medium-sized enterprise (SMEs) [49]. Collaboration among employers, employees, and society is essential to prioritize employee health in the workplace [46]. Additional implementation barriers include discrepancies between coaches’ assessments and exercisers’ perceptions [48], technological limitations [41], and remote work-related factors such as reduced peer support, stress management, and sleep quality challenges [47].

Future WPABPs must address long-term adherence, particularly in the post Coronavirus Disease-2019 (COVID-19) context [47,57], while balancing cost, scalability, and individual needs. High-intensity, long-duration, and group-based programs tend to generate more sustained improvements, whereas low-intensity or short-term interventions often yield transient effects [12,19,44]. Group interventions support social cohesion and organizational culture, while individualized coaching better addresses stress, burnout, and high-pressure roles [48]. Digital and tele-exercise tools can extend reach but risk information overload and superficial engagement if intrinsic motivation is neglected [41,51,58]. Societal shifts, including aging populations, urbanization, and remote work, alongside organizational priorities favoring short-term returns, further complicate implementation, highlighting the need for context-sensitive, flexible, and resource-conscious strategies to maximize employee psychological and organizational benefits [2,56,58].

### 4.6. Workplace Accessibility and Equity Issues

This review reveals that organizations capable of implementing the WPABPs are typically large enterprises, public institutions, or organizations with established human resource and health promotion systems. For instance, the UK NHS Time to Move initiative demonstrated that when employers provide paid exercise time, employees’ participation, mental health, and job satisfaction significantly improve [42]. In contrast, SMEs, outsourcing firms, and shift-based or hourly workers—such as those in the BPO industry—often lack the financial, temporal, or managerial capacity to implement high-quality or individualized interventions [12,46]. Moreover, much of the current evidence is drawn from high-income countries and office-based sectors, resulting in limited representation of workers in industrial, manual, or service-oriented occupations [19,43,44,45].

Differences in occupational health and safety (OHS) policy environments significantly affect accessibility to the WPABPs. In countries with well-developed OHS frameworks—such as the United Kingdom, Germany, Sweden, Australia, New Zealand, and Spain—WPABPs are more easily integrated into organizational structures and often supported by systemic initiatives [12,21,41,42,43,50,51]. Public healthcare systems (e.g., NHS staff in Wales) and higher-education institutions (e.g., Spanish universities) provide structured programs and organizational support, enabling employees to participate in physical and psychological health interventions with measurable benefits for work satisfaction, mental health, and productivity [19,49]. Countries with moderately strong OHS policies, such as Portugal, provide partial support for WPABPs, often relying on organizational or individual initiative [48]. Conversely, in countries with weaker or less formalized OHS environments—such as Brazil and some African nations—the integration of WPABPs is limited, reducing opportunities for employee engagement in health-promoting activities [19]. Remote and desk-based workers, common in both high- and low-resource settings, face additional challenges due to sedentary behavior, reduced social support, and elevated stress [12,47]. Tailored interventions are therefore necessary even in countries with strong OHS policies. Overall, the accessibility and effectiveness of WPABPs are closely linked to national OHS frameworks, with stronger policies facilitating implementation and participation, while weaker infrastructures disproportionately affect vulnerable populations.

In summary, both policymakers and corporate executives are encouraged to take joint responsibility to establish inclusive, flexible, and sustainable workplace physical activity policies. Governments need to provide necessary technical and financial support to SMEs, while companies must enhance managerial awareness of health promotion as part of ESG-S and sustainability strategies. Only through institutionalized, multi-level, and cross-departmental collaboration can WPABPs truly become embedded in organizational culture, achieving a win-win for corporate competitiveness and employee well-being.

### 4.7. Disparities in Participation: Socioeconomic and Occupational Gaps in WPABP Research

A critical gap in the current WPABP literature concerns disparities in access to workplace health promotion across industries, organizational sizes, and socioeconomic contexts. Most empirical studies were conducted in developed economies—primarily in Europe, the UK, and Australia—within public institutions or large organizations possessing strong OHS infrastructures [21,42,43]. By contrast, SMEs, labor-intensive sectors, and informal work environments remain underrepresented, despite facing greater health risks and limited structural support [44,46]. Furthermore, existing research tends to emphasize white-collar and sedentary occupations, overlooking blue-collar or shift-based workers who may experience higher occupational strain and fewer opportunities for structured health programs [45,52].

Future research should therefore expand to diverse socioeconomic and occupational contexts, integrating intersectional analyses of gender, job type, and employment status to better capture inequities in participation and outcomes. Methodologically, participatory and community-based approaches could improve inclusiveness and ecological validity, particularly in low-resource workplaces. Addressing systemic barriers and fostering supportive workplaces are essential to improving essential workers’ health, well-being, and efficiency. Involving personal support workers in decision-making and prioritizing resources (e.g., healthcare benefits, mental health support) can create healthier work environments [59]. Policy-level collaboration is also needed to embed equitable access to physical activity programs within OHS frameworks, ensuring that all employees—not only those in well-resourced organizations—can benefit from workplace health promotion [12,41,47,60].

## 5. Theoretical Implications

Grounded in the S–O–R framework, this review advances the theoretical understanding of how WPABPs serve as environmental stimuli that shape employees’ internal psychological states (organism) and subsequent behavioral and attitudinal outcomes (responses) [29]. The integration of the S–O–R model provides a dynamic lens through which the mechanisms linking workplace interventions and sustainable Human Resource Management (HRM) can be interpreted (Figure 2).

### 5.1. WPABPs as Environmental Stimuli in Sustainable Work Environments

From the S–O–R perspective, WPABPs constitute structured stimuli within the organizational environment. These programs act as tangible manifestations of an organization’s commitment to employee welfare under the ESG-S. The findings indicate that both individualized and group-based exercise programs stimulate a positive organizational climate—providing physical spaces, social interactions, and managerial encouragement that elicit employees’ attention and emotional engagement [19,42,43]. This extends the S–O–R model by contextualizing stimuli not merely as sensory or environmental cues, but as strategic HRM interventions that embody corporate sustainability and social responsibility [45,51].

### 5.2. Psychological Processes as Organismic Mediators

The organism component is represented by employees’ psychological states. Consistent with self-determination theory and S–O–R logic, WPABPs foster psychological need satisfaction, reduce emotional exhaustion, and enhance vitality [12,48]. The review identifies positive affect, mindfulness, and stress recovery as critical mediators between workplace exercise stimuli and outcomes such as mental health, motivation, and engagement [41,47]. This expands the traditional S–O–R understanding by integrating psychological capital and well-being mechanisms as internal organismic states linking HRM interventions to sustainable behavioral outcomes [21,49].

### 5.3. Behavioral and Attitudinal Responses as Sustainability Outcomes

The response dimension encompasses employees’ behavioral changes (e.g., exercise adherence, reduced sedentary behavior, improved performance) and attitudinal outcomes (e.g., job satisfaction, organizational commitment, and social cohesion). By highlighting the consistent positive associations between WPABPs and work-related outcomes (e.g., productivity, teamwork, and culture formation). This review extends the S–O–R framework toward organizational sustainability outcomes [43,44,50]. In doing so, it bridges the gap between individual-level behavioral responses and collective-level sustainability practices in corporate contexts [19,42].

### 5.4. Differentiation Between Individual and Group Pathways

The comparison between individualized and group interventions introduces a du-al-pathway interpretation within the S–O–R model. Individualized stimuli (e.g., trainer-led or tele-coaching programs) primarily influence deep psychological states—stress reduction and personal satisfaction [47,49]—whereas group-based stimuli reinforce social cohesion and collective engagement [21,48]. This dual-pathway structure enriches S–O–R theory by demonstrating that stimulus configuration and delivery mode moderate the strength and type of organismic and response effects, suggesting a nuanced, multilevel interpretation of S–O–R processes in workplace settings [12].

### 5.5. Embedding S–O–R in the ESG-S and Sustainable HRM Context

The synthesis of S–O–R with the ESG-S framework contributes to theoretical integration between organizational psychology and sustainability science. WPABPs operationalize ESG-S by demonstrating how psychological well-being functions as a bridge between CSR (stimulus), employee mental states (organism), and long-term sustainable behaviors (response). Thus, this review positions S–O–R as a conceptual anchor for sustainable HRM, linking workplace health promotion with broader social sustainability and human capital development goals SDG 3, 5, and 8 [43,45].

### 5.6. Toward a Multi-Level S–O–R Perspective

Finally, the findings underscore the need to extend the traditional individual-level S–O–R model to a multi-level organizational framework. The effectiveness of WPABPs depends on managerial support, organizational culture, and policy structures—representing macro-level stimuli that interact with micro-level organismic processes [42,43]. This multi-level view encourages future theoretical models to incorporate feedback loops—where improved employee well-being (response) reinforces sustainable organizational practices, creating a self-sustaining cycle of social responsibility and human resource sustainability [19,50].

In summary, this theoretical integration situates WPABPs within the broader agenda of sustainable workplace transformation, thereby enhancing the explanatory power of S–O–R for future ESG-S-driven behavioral research.

### 5.7. Practical Implications

The findings of this review provide several important implications for managers, HR practitioners, and organizational decision-makers seeking to enhance employee well-being and sustainable workplace performance through WPABPs.

First, managers should recognize that program design must be context-specific [45,56]. The considerable heterogeneity in intervention types, intensities, and delivery formats across occupational and national settings indicates that a one-size-fits-all approach is unlikely to be effective. WPABPs should prioritize underrepresented groups, particularly sedentary and resource-constrained employees who face barriers to participation [41,46]. Organizations should assess employee needs, job demands, and workplace characteristics when selecting or designing interventions, particularly in diverse sectors such as healthcare, banking, and business process outsourcing (BPO), where risk profiles and resource availability differ substantially.

Second, managerial support and organizational culture emerged as critical drivers of program success. Evidence consistently showed that leadership endorsement, flexible scheduling, and supportive workplace norms substantially increased participation, motivation, and mental health outcomes. Managers can enhance program effectiveness by formally allocating time for physical activity (e.g., paid activity hours), reducing structural barriers, and modeling supportive behaviors. Institutional or governmental support, especially for SMEs, can further mitigate financial and infrastructural limitations [44,46]. Ensuring that all employees, regardless of resources, benefit from WPABPs not only promotes fairness but also enhances engagement, program effectiveness, and long-term adherence.

Third, the review highlights the value of hybrid and flexible implementation strategies. Combining group-based activities with individualized or trainer-led components can address both scalability and personalization needs. Hybrid or group-based models can expand reach and improve resource efficiency [12], while flexible scheduling, paid exercise time, and remote participation options help reduce structural obstacles [41]. Integrating digital tools may support remote or desk-based workers, although careful consideration of technology overload and motivational barriers is required.

Fourth, organizations should leverage interpersonal and psychosocial mechanisms, such as supportive coaching, positive affect, and social cohesion, to strengthen intrinsic motivation and sustain participation. Investing in qualified trainers or wellness facilitators who understand psychological need alignment can enhance both employee engagement and workplace atmosphere.

Finally, adopting WPABPs contributes directly to ESG-S and Corporate Social Responsibility goals, offering measurable social sustainability outcomes, including reduced stress, improved mental well-being, increased productivity, and potential economic benefits. Managers aiming to strengthen ESG reporting and human capital development should integrate WPABPs into broader sustainability strategies and occupational health policies.

These implications suggest that organizations can maximize the mental health and performance benefits of WPABPs by prioritizing contextual tailoring, managerial support, accessibility, and long-term engagement, ultimately fostering healthier, more resilient, and socially responsible workplaces. Meanwhile, incorporating “accessible health and well-being resources” as a measurable indicator within OHS and ESG-S frameworks can further institutionalize equity in workplace health policies.

## 6. Conclusions and Future Research

This systematic review demonstrates that WPABPs function as structured environmental stimuli that positively influence employees’ psychological, physical, and social well-being, which in turn drive beneficial behavioral and organizational outcomes. By integrating the S–O–R framework with ESG-S and CSR principles, WPABPs are shown to enhance mental health, stress recovery, vitality, social cohesion, teamwork, organizational culture, and productivity. Individualized interventions primarily support deep psychological gains and targeted health outcomes, whereas group-based programs reinforce collective engagement, motivation, and organizational culture. A hybrid approach combining both individual and group modalities appears most effective in balancing personalization, scalability, and inclusiveness.

Despite consistent evidence of positive effects, significant heterogeneity exists in intervention types, intensity, duration, and delivery methods. Most studies rely on self-reported data, are cross-sectional, and focus on office-based or healthcare employees in high-income countries, limiting generalizability [12,44,45]. Frontline industrial workers, small enterprise staff, part-time or shift employees, and those in low- and middle-income contexts remain underrepresented, highlighting systemic inequities in access to workplace health promotion [46,47].

Future research should standardize intervention protocols and adopt longitudinal or experimental designs to establish causal relationships. Expanding sampling to diverse occupations, socioeconomic contexts, and underrepresented employee groups will improve equity and generalizability. Evaluations that integrate economic, psychological, and organizational outcomes can provide robust evidence of cost-effectiveness and sustainability. Additionally, comparative and cross-cultural studies are needed to inform context-sensitive and culturally relevant workplace health policies.

In conclusion, WPABPs represent a high-impact, sustainable strategy that bridges employee well-being with organizational performance. When thoughtfully designed, effectively implemented, and institutionalized, these programs can transform isolated health interventions into integrated, long-term mechanisms that enhance both individual well-being and organizational sustainability. This review highlights the potential of WPABPs to serve as a practical roadmap for future research and real-world applications across diverse workplace settings.

## Figures and Tables

**Figure 1 healthcare-13-03146-f001:**
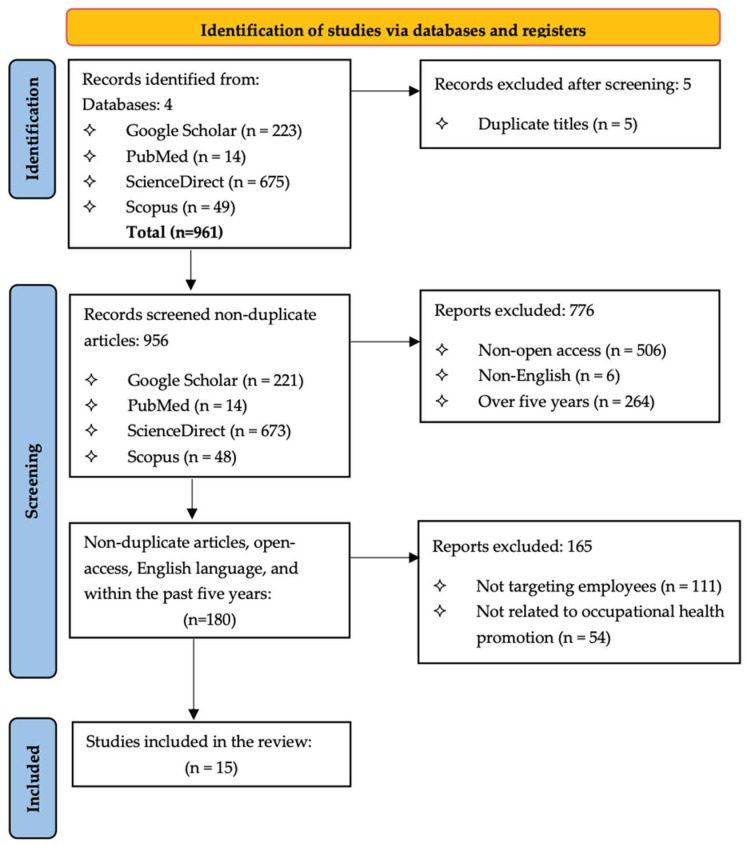
Flowchart of the screening process and selection criteria.

**Figure 2 healthcare-13-03146-f002:**
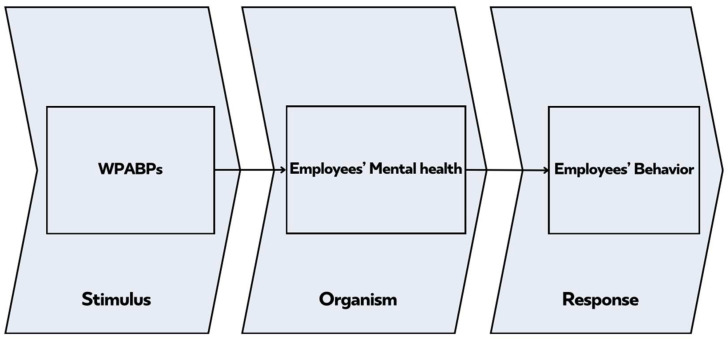
Theoretical Framework.

**Table 1 healthcare-13-03146-t001:** Studies Applying the S–O–R Framework to Workplace Employees.

Author (Year)	Title	Stimuli	Organism	Response	**Operationalization of Variables**
Kılıç et al. (2025) [30]	Do Green Hotels Lead Employees to Green Behavior? Application of Stimulus-Organism-Response Theory and Organismic-Integration-Theory	Green hotel practices (e.g., green policies, eco-friendly operations).	Employees’ internal psychological states regarding environmental values and motivation.	Green behaviors performed by employees in the workplace.	Stimuli measured through green hotel practice scales; organism via environmental motivation and internalization constructs; responses via employee green behavior scales.
Taufik et al. (2025) [31]	Enhancing Worker Productivity through the S–O–R Theory in Human Resource Management	HRM-related stimuli such as rewards, leadership, and work environment.	Employees’ psychological reactions, including motivation and job attitudes.	Worker productivity and performance outcomes.	Stimuli operationalized through HRM practice indicators; organism via motivation/attitude scales; responses via productivity self-report and supervisor ratings.
Çelik et al. (2024) [32]	Examining Employees’ Job Satisfaction and Life Satisfaction in the Context of the S–O–R Model	Workplace conditions and job-related stimuli (e.g., work environment, organizational policies).	Employees’ internal evaluations of job satisfaction and life satisfaction.	Behavioral and attitudinal outcomes linked to satisfaction levels.	Workplace stimuli measured via organizational climate scales; organism via job satisfaction and life satisfaction scales; responses via behavioral intention measures.
Jiang et al. (2022) [33]	How Smart Technology Affects the Well-Being and Supportive Learning Performance of Logistics Employees?	Stimuli derived from smart technology use and digital systems in logistics workplaces.	Psychological well-being, perceived support, and cognitive-emotional states of employees.	Supportive learning performance and employee outcomes.	Smart technology stimuli measured through technology acceptance/usage scales; organism via well-being and perceived support metrics; responses via learning performance indicators.
Tang et al. (2019) [34]	Understanding employees’ energy saving behavior from the perspective of stimulus-organism-responses	Organizational energy-saving cues, policies, and environmental prompts.	Employees’ internal cognitive states such as environmental concern and perceived responsibility.	Energy-saving behavior at work.	Stimuli assessed using organizational energy policy and cue scales; organism via environmental cognition and responsibility constructs; responses via self-reported energy-saving behaviors.
Chung et al. (2024) [35]	Influence of Corporate Social Responsibility on Employees’ Creative Behavior and Turnover Intention in Professional Team Sports Organizations: The Mediating Role of Subjective Well-Being	Corporate Social Responsibility initiatives perceived by employees.	Employees’ subjective well-being as an internal psychological state.	Creative behavior and turnover intention.	CSR stimuli measured through CSR perception scales; organism via subjective well-being scales; responses via creative behavior and turnover intention scales.

Note: Human Resource Management (HRM); Corporate Social Responsibility (CSR); Stimulus–Organism–Response (S–O–R).

**Table 2 healthcare-13-03146-t002:** Mapping of the Selected Keywords to MeSH Thesaurus Terms.

Keywords	
MeSH Entry Term (Equivalent/Synonymous)	MeSH Preferred Term
workplace/worksite	workplace
employees/workers	personnel/occupational groups
physical activity	exercise
wellness program/health Programs	health promotion
well-being/mental health	mental health

**Table 3 healthcare-13-03146-t003:** Inclusion and Exclusion Criteria.

Category	Inclusion Criteria	Exclusion Criteria
Population	Employees working in organizational or occupational settings across sectors	Non-employees (e.g., students, unemployed individuals, or non-workplace-related)
Intervention	Interventions involving any form of occupational health promotion	Interventions unrelated to occupational health promotion
Comparator	Studies including a control, comparison, or pre-post design examining the effects of workplace exercise interventions	Studies lacking a comparison group or not assessing the effects of exercise interventions
Outcome	Studies reporting measurable or reported outcomes related to physical and mental health, or work-related outcomes	Studies not reporting measurable or relevant health or work-related outcomes
Study design	Clearly described methodology; original research articles, RCTs, observational studies, systematic reviews, meta-analyses, quasi-experimental studies, narrative reviews	Unclear methodology; experimental abstracts, case reports, editorials, letters to the editor
Other	Non-duplicate records; full-text articles availability; English-language publications; published within the past five years	Duplicate records; articles without full text; non-English-language publications; published more than five years ago

**Table 4 healthcare-13-03146-t004:** Integration of Included Articles from Each Database.

Database	ScienceDirect	PubMed	Scopus	Google Scholar
Selected Articles	3	7	2	3
Total Articles	15			

## Data Availability

No new data were created or analyzed in this study.

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
