# Peer review of "Sustainable Workplaces and Employee Well-Being: A Systematic Review of ESG-Linked Physical Activity Programs"

_healthcare, 2025, doi:10.3390/healthcare13233146_

Round 1
Reviewer 1 Report
Comments and Suggestions for Authors
In this manuscript, the authors report the results of a systematic review examining the impact of Workplace Physical Activity-Based Programs (WPABPs) on employees’ mental health.
The manuscript is clear and well-written, I also appreciate that the authors followed the PRISMA guidelines for reporting systematic reviews. The results are sound and follow the findings of the systematic review. I have just two very-minor comments for the authors:
- why non-open access papers have been excluded from the systematic review?
- rows 81-84: "In summary, the WBPABPs not only help enhance employees’ physical and mental health and well-being but also promote productivity, job satisfaction, and organizational loyalty. By combining individual health promotion and work performance improvement, physical activity programs have become important strategies for strengthening employee well-being and building sustainable competitive advantages for companies." This section, in the introduction, is speculative, since you have already to demonstrate these findings. So I suggest using the conditional (e.g., "may not only help...").
Author Response
Please find the attached document for your review.

Reviewer 2 Report
Comments and Suggestions for Authors
The manuscript brings the current topic and is excellently written and structured. The manuscript has many strengths, but there are a few weaknesses that could be improved.
1. The abstract is quite informative, but there are elements that need to be revised. The results are described in one sentence, and the conclusion makes up the majority of the text in the abstract. This ratio is not good. Also, the text contained in the section related to the conclusion is actually the results. It is necessary to revise the abstract so that a clear delineation is made between the results and the conclusion, and in particular it is necessary to take into account what the results of the study are.
2. The Introduction section is well structured and written. The research objective and purpose are clearly defined.
3. The research method is described in a separate section and presented graphically. Figure 1 essentially provides an overview of the methods and should therefore be positioned at the end of the Method section, rather than the Results section.
4. The discussion is related to the results of the study, it is sufficiently informative and clear. The following theoretical implications provide good support for the research results.
5. The last section is well designed and contains limitations of the study, as well as future research directions. Despite the fact that the work is not based on empirical research, the obtained results need to be put into the context of practice. Therefore, it is recommended to add practical implications in this part, which will, on the one hand, summarize the results of previous research (previous studies that are included in this research), and on the other hand, synthesize them and formulate practical implications based on that.
Author Response

(The authors gave the same response as above.)

Reviewer 3 Report
Comments and Suggestions for Authors
Dear authors,
The topic is relevant, and I commend the authors' efforts; however, some clarifications and corrections are required to improve the manuscript.
Below are the key areas in which the manuscript can be improved:
- Abstract
- The abstract is well-structured. However, the aim/objective of the research should not precede the background because doing so reverses the logical flow of the study. Moreover, readers first need to understand the existing knowledge base, unresolved problems, or knowledge gaps before they can appreciate why the specific is necessary and justified. Revise according.
- Introduction
- On page 3, section 1.4, the discussion of SOR theory and its applicability to the topic requires more development. Currently, SOR is mentioned only briefly and superficially without a systematic examination of how the included studies operationalize stimuli, organismic, and responses. As it stands, the treatment of SOR misses the opportunity to provide deeper conceptual insights.
- The authors should create a subsection that synthesizes and critically assesses the aforementioned comment. Specifically, create a summary Table mapping SOR components across various studies to enhance theoretical coherence and contribution of the review
- Methodology
- The methodology and procedure section of the manuscript is well written and structured in the research approach, study search, and inclusion criteria. However, the data extraction process could be further developed. Were the records screened for duplications? Were the retained articles thoroughly read? Is the PRISMA flow diagram accurate and complete, with the exact number of records excluded at each phase and the reason for full-text exclusion?
Without this information and further developing this section of the manuscript, the risk of bias cannot be addressed, and the reliability of the entire review is compromised.
- Conclusion
This section of the manuscript needs further development. The conclusion section is overly repetitive and brief. To improve it, clearly delineate how the current manuscript advances the existing theory and conclude with a closing sentence that underscores the study's significance rather than fading out.
Author Response

(The authors gave the same response as above.)
